# Novel Experimentation for the Validation of Mechanistic Models to Describe Cold Dwell Sensitivity in Titanium Alloys

**Elizabeth E. Sackett [1],\* and Martin R. Bache [2]** 

1   Faculty of Science and Engineering, Swansea University, Swansea SA1 8EN, UK
2   Institute of Structural Materials, Swansea University, Swansea SA1 8EN, UK; m.r.bache@swansea.ac.uk
\*   Correspondence: e.sackett@swansea.ac.uk; Tel.: +44-(0)1792-295413

**Abstract:** Previous mechanistic models, proposed to explain the process of damage accumulation and stress redistribution between strong and weak regions inherent within the microstructure of $\alpha/\beta$ and near $\alpha$ titanium alloys, are validated through a matrix of experiments employing a non-standard variant of the alloy Ti 685. The grain size of the model material was deliberately processed to offer grains up to 20 mm in diameter, to facilitate constitutive measurements within individual grains. A range of experiments were performed under static and cyclic loading, with the fatigue cycle conducted under either strain or load control. Data will be reported to demonstrate significant variations in elastic and plastic properties between grains and emphasise the role of time dependent strain accumulation. Implications for the "dwell sensitive fatigue" or "cold creep" response of conventional titanium alloys will be discussed.

**Keywords:** dwell fatigue; cold creep; microstructurally textured regions; titanium



## 1. Introduction

Ever since the seminal work of Evans and Gostelow in the early 1970s [1], resulting from the investigation of in-service fan disc failures in gas turbine engines powering the Lockheed Tristar [2], key factors associated with "dwell sensitive fatigue" fractures were identified as:

- Stress redistribution occurring between weak and strong microstructural regions
- An increased rate of cyclic strain accumulation under fatigue cycles incorporating a hold time at peak or high mean stress
- Cracks were initiated through the formation of quasi-cleavage facets orientated orthogonal to the applied tensile stress axis and often sub-surface

The specific alloy of interest during that period was the near $\alpha$ alloy Ti 685, originally developed for high temperature capability but subsequently employed throughout various fan-compressor stages. The relative differences in constitutive strength were offered by neighbouring hexagonal grains, of the order of 300 µm diameter, arranged with random crystallographic orientations. Alternatively, variations in the post forged microstructure were noted, notably undesirable regions of aligned alpha laths within the favoured basketweave structure [3]. Alloy developments led to alternative near $\alpha$ alloys for compressor applications, for example Ti 834 and Ti 6242. Although optimised for elevated temperature employment, neither was able to avoid the sensitivity to dwell fatigue at room temperature [4,5].

For fan disc components, the major gas turbine manufacturers have since relied upon the $\alpha/\beta$ alloy Ti-6Al-4V on the understanding that annealed and bi-modal microstructural variants of this alloy were essentially dwell insensitive. A recent investigation of an in-service fan disc failure has concluded, however, that dwell fatigue failures may be initiated in Ti-6Al-4V components subject to the presence of "macrozones" or "microstructurally

textured regions" (MTRs) [6]. That conclusion is consistent with the outputs from a contemporary, multi-national study active over the past five years (for example see reference [7]). The Materials Affordability Initiative (MAI) project, sponsored by the US Department of Defense with collaboration between the lead gas turbine manufacturers, academia and government research institutions, has attempted to identify examples of MTRs in forged Ti-6Al-4V materials and develop bespoke inspection techniques to characterize their size and crystallographic form as a pre-requisite to advanced computer modelling [8]. In parallel to the MAI consortium, a number of additional workers have recently published evidence of cold dwell behaviour in Ti-6Al-4V (for example [9–11]). MTRs represent regions within the microstructure where the hexagonal $\alpha$ phase sub-elements (either retained as primary $\alpha$ grains or transformed back to $\alpha$ from the $\beta$ phase during thermo-mechanical heat treatment) have similar crystallographic orientation. Under applied stress, depending on the orientation of the individual hexagonal units relative to the principal loading axis, these MTRs may act as a single large microstructural unit. It is proposed that if adversely orientated amongst relatively weaker regions of the structure the strong MTRs can encourage extensive zones of quasi-cleavage facets to form and promote the transition into fatigue crack growth [12].

Based on an extensive body of work on Ti 685 in particular, Bache proposed a two-element model to describe the accumulation of local stresses and strains between neighbouring sub-surface (constrained) grains and resultant stress redistribution that may occur between adjacent "weak" and "strong" grains [13]. Invoking the Stroh model [14], Evans and Bache previously promoted this process to explain the formation of quasi-cleavage facets, reliant on time dependent planar slip, dislocation pile up and progressive opening of a facet on the basal plane of the neighbouring "strong" grain [15]. Subsequent studies of finer grained near $\alpha$ and $\alpha/\beta$ variants containing MTRs has recently led to a modification of the Evans–Bache model to account for highly textured microstructures on a regional scale [16].

Mechanical studies to validate these models often resort to the use of sophisticated experimental techniques applied to standard microstructural variants to measure localised damage on the length scale of individual grains, for example, the use of acoustic emission to detect sub-surface crack initiation events [5] or back scattered electron diffraction and digital image correlation to correlate cracks with grain orientation [17]. The experiments described here adopted an alternative, novel approach. A bespoke high vacuum heat treatment routine was applied to standard Ti 685 to deliberately grow the grain size to a scale where conventional laboratory characterisation techniques could be employed within individual grains. Room temperature "cold creep" experiments were performed under static load to monitor bulk deformation and the differing elastic–plastic contributions within individual grains. Load and strain-controlled fatigue assessments generated fatigue life data and examples of the faceting mechanism on a scale that could be observed by eye. These combined loading scenarios provided evidence of the cyclic softening behaviour commonly observed in these materials under both unconstrained and constrained conditions. A distinct difference between the monotonic and cyclic stress–strain relationship is illustrated and the fundamental link between the occurrence of a dwell fatigue debit and cyclic yield behaviour is proposed. Whilst offering a qualitative validation of previous theoretical models that describe the cold dwell phenomenon, the present data could also form the basis for computer-based crystal plasticity, for example [18].

Differences in plastic behaviour on the grain scale can be detected by eye on the surface of mechanically turned or milled parts prior to final polishing or chemical treatment. Commercially pure titanium [19] and from personal experience conventional forms of near $\alpha$ titanium alloys with a relatively large grain size such as Ti 829 and Ti 685 (of the order 0.25 mm ferret diameter) often generate an "orange peel" effect at their surface, controlled by crystal orientation, grain rotations and the propensity or otherwise to induce transgranular planar slip [19]. It is proposed that while the present investigation aims to address sub-surface grain to grain interactions and specifically the cold dwell effect, at the

same time it would provide constitutive measurements that shed light on and help such model surface phenomena.

## 2. Materials and Methods

A novel, large grained form of the near α titanium alloy Ti 685 (Ti-6Al-5Zr-0.5Mo-0.25Si) was employed for the basis of this investigation. Rectilinear blocks of conventional material with nominal dimensions 180 × 180 × 25 mm were subjected to a supplementary proprietary vacuum heat treatment. This produced an average grain diameter of 15 mm. On occasions grains approaching 20 mm diameter were noted. Post heat treatment, the grains were clearly visible to the naked eye without the need for chemical etching (Figure 1). Metallographic inspection confirmed a coarse, interlocking, primary α lath structure within the individual grains. This "model" material has been designated LG685.

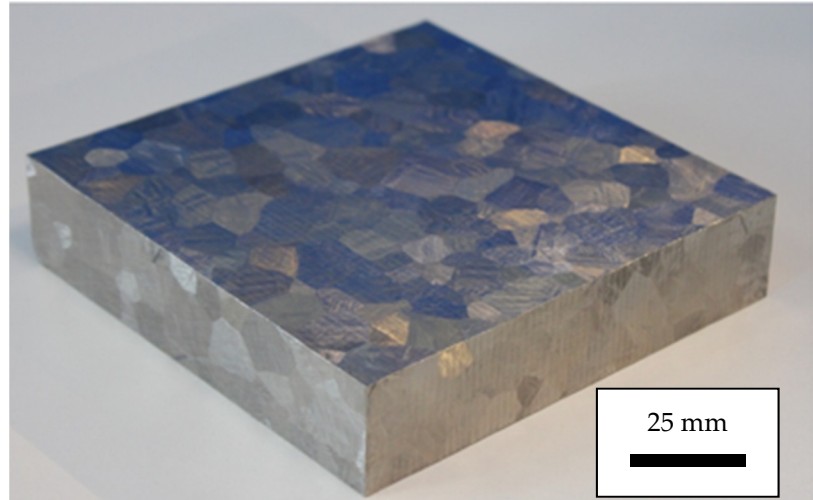

**Figure 1.** Grain structure revealed at the surface of a vacuum heat-treated block of LG685.

Two specimen geometries were adopted for various forms of static and cyclic mechanical testing, a plain cylindrical specimen and a flat plate geometry, illustrated in Figures 2 and 3 respectively.

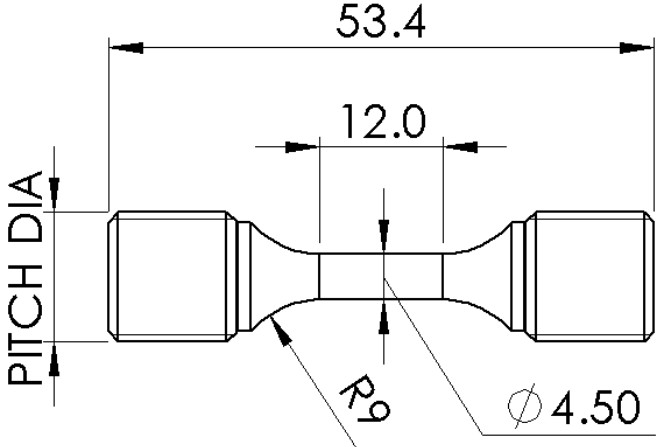

**Figure 2.** Round cylindrical specimen geometry (dimensions in mm).

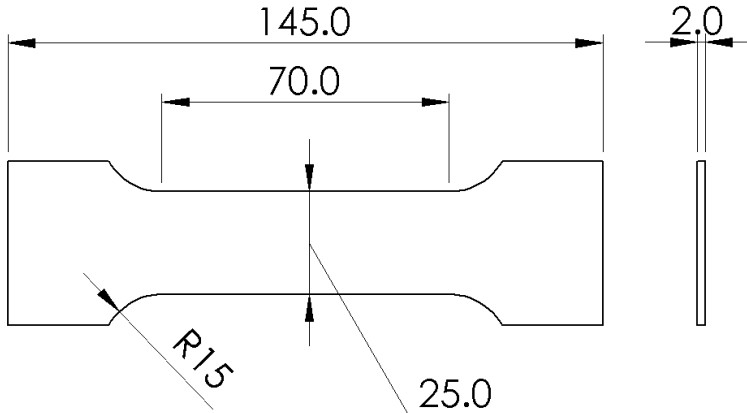

**Figure 3.** Flat plate specimen geometry (dimensions in mm).

It is recognised that from the perspective of materials testing standards neither design would qualify as suitable test pieces for characterising bulk materials response, given that the number of grains sampled within the critically stressed volume of either geometry was not sufficient to measure average or "bulk" constitutive behaviour. As a consequence of the relatively large grain size, the gauge section of the cylindrical specimens in particular could feasibly sample just a single grain (it should be emphasized that all specimens were extracted from random locations within the plates). However, as will be demonstrated, they served the objectives of the current investigation, to characterise different properties in adjacent grains of varying crystallographic orientation.

The flat plate design was employed for static loading experiments conducted under a linear load-controlled ramp at 1 mm per minute (1.666 ms$^{-1}$). Specimens were either loaded to induce complete rupture or loaded to a pre-determined load and then held over a period of time. The as machined surface finish to these specimens (milled and lightly ground) allowed the grain structure to be defined by eye on the opposing flat surfaces. Multiple, strain gauges were applied using a proprietary adhesive (Vishay Precision Group—Micro Measurements, Toronto, ON, Canada) in accordance to B-127 [20] to the central region of randomly selected, individual grains as illustrated in Figure 4. The area of the electronic gauge rosette was 15 mm$^2$. Constitutive data were recorded in the form of machine load, machine position and strain gauge displacement at a sample rate of 25 Hz.

Plain cylindrical specimens were employed for room temperature constant amplitude strain controlled LCF testing to establish fatigue strength ($\varepsilon$-N data), constitutive stress–strain hysteresis loop information ($\sigma$-$\varepsilon$) and generate freely initiated fractures for inspection. Strain control fatigue was carried out according to BS7270. The applied strain cycles were controlled by a side mounted strain gauge bridge extensometer set to a 10 mm gauge length. A trapezoidal waveform (15 cycles per minute, comprising 1 s rise and fall ramps with a 1 s hold at both peak and minimum strain) was employed at R = 0.1, with peak strain values selected in order to induce fatigue failures in less than $10^5$ cycles. As per BS7270, failure was determined once a 10% drop was noted from the stabilised stress condition.

Cylindrical specimens were also employed for constant amplitude load controlled LCF testing at room temperature which were conducted in accordance to BS3518 to establish baseline and dwell fatigue strength ($\sigma$-N) and to generate freely initiated fractures for inspection. "Cyclic" fatigue loading was applied utilizing a 15 cycle per minute trapezoidal waveform at R = 0.1, with peak stresses selected to induce fatigue failures in less than $10^5$ cycles. The "dwell" fatigue tests employed identical 1 s loading and unloading linear ramps, a 1 s hold at minimum load (stress), but with a 2 min hold imposed at the peak of the waveform. The two distinct waveforms are illustrated schematically in Figure 5.

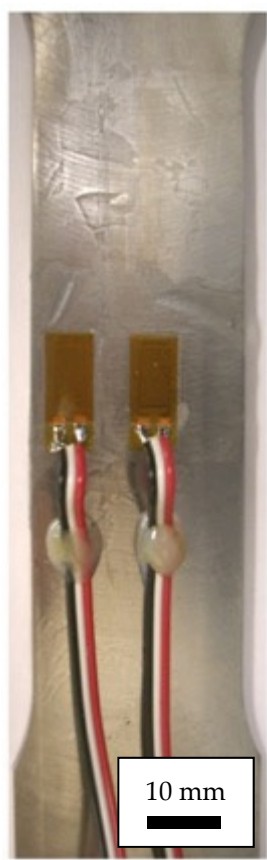

**Figure 4.** Strain gauges applied within adjacent grains in a flat plate specimen.

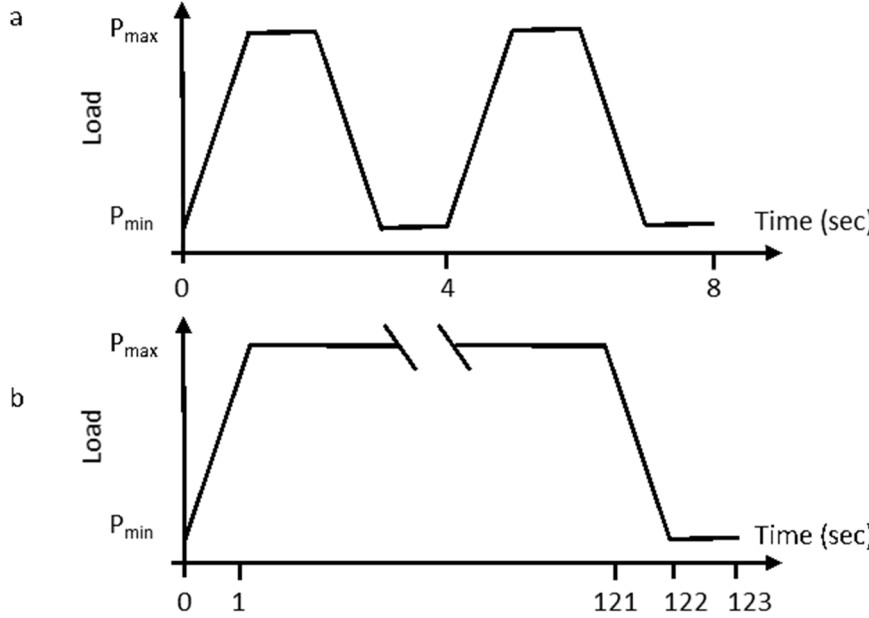

**Figure 5.** Schematic representation of (**a**) cyclic and (**b**) dwell load controlled fatigue waveforms.

Stress relaxation tests also employed the plain cylindrical specimen design. The specimen was loaded under an extension-controlled ramp (controlled via the side mounted extensometer) to apply a specific value of strain and then held at this strain for a period of 24 h. During the hold period continuous measurements of resultant load were recorded.

## 3. Results

### 3.1. Monotonic and Static Behaviour

A stress–strain curve measured from a single flat plate specimen is presented as Figure 6. The measured UTS at 768 MPa and approximate yield stress at 600 MPa (taken as the point of deviation from the proportional elastic response) display a lower strength for this large-grained variant compared to conventional Ti 685 materials with either basketweave or aligned microstructure, as previously evaluated in this laboratory [21]. This equates to a strength debit around 20%.

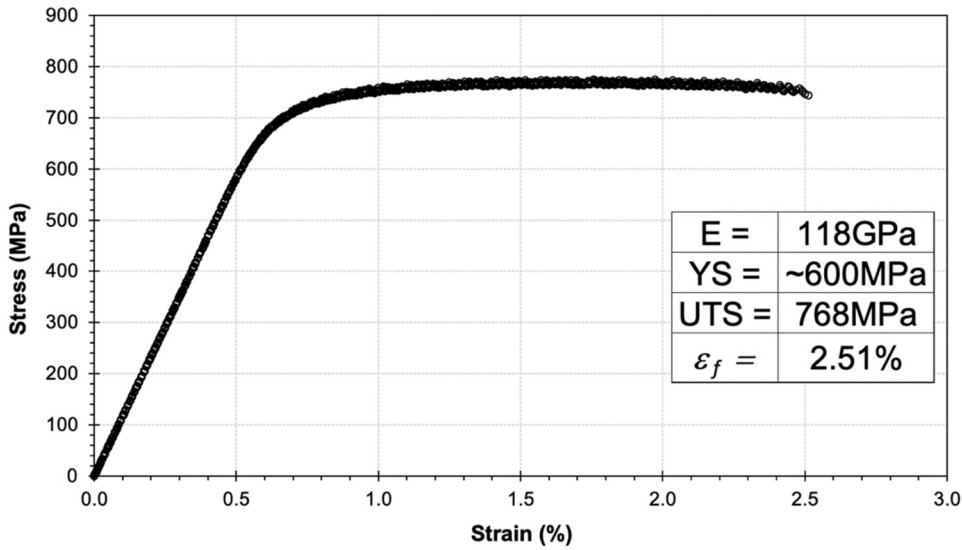

**Figure 6.** Stress–strain curve measured from a flat plate LG685 specimen.

This appears consistent with classical relationships between strength and grain size, e.g., the Hall–Petch relationship. Repeat tests of this nature were prohibited due to the volume of material available; however, it is probable that all of the basic mechanical parameters would have been subject to statistically significant scatter dependent on the number and crystallographic orientation of the grains sampled within individual specimens. This assumption was supported by two independent sets of data. Firstly, Vickers hardness measurements (taken to represent yield behaviour) as sampled by a single indentation at the centre of seven neighbouring grains along a randomly drawn line on a single machined material face, Table 1. Secondly, measurements of the elastic modulus taken from six individual grains within multiple flat plate specimens via strain gauge, Table 2.

**Table 1.** Vickers hardness measurements from the centre of seven neighbouring grains (single indentations).

| Grain I.D. | Hardness ($H_V$) |
|:---:|:---:|
| 1 | 225 |
| 2 | 235 |
| 3 | 358 |
| 4 | 384 |
| 5 | 367 |
| 6 | 371 |
| 7 | 307 |

**Table 2.** Elastic modulus from six individual grains measured by strain gauge.

| Grain I.D. | E (GPa) |
|---|---|
| 1 | 115 |
| 2 | 109 |
| 3 | 109 |
| 4 | 102 |
| 5 | 106 |
| 6 | 103 |

Two individual grains randomly selected within a single flat plate specimen and sampled concurrently via strain gauges illustrated different time dependent deformation under static loading at 600 MPa, Figure 7.

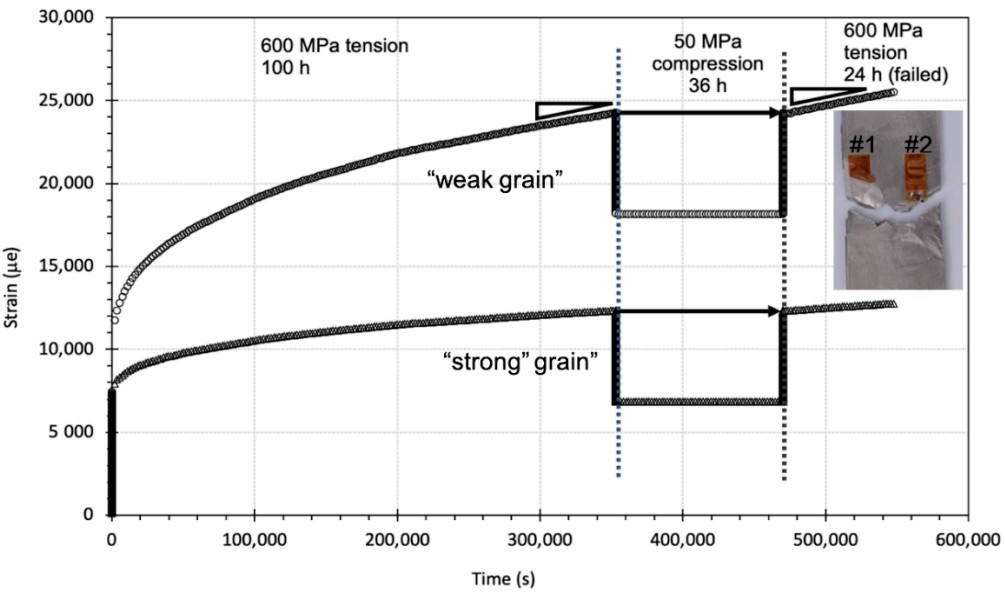

**Figure 7.** Strain accumulation over time within two individual grains.

Essentially, this can be considered as a room temperature or "cold" creep test. Immediately following loading the rate of primary strain accumulation in grain G#2 clearly exceeds that of grain G#1. The differences in strain rate persist into the steady state regime. After 100 h the load was reversed into a slight compression of 50 MPa for a period of 36 h. Permanent but constant strain damage was measured throughout the compressive period. Then the tensile stress of 600 MPa was reapplied, to recover a tensile strain at an identical magnitude prior to the load reversal in either grain. Subsequent strain accumulation was measured at an identical steady state rate. Ultimately, the specimen failed after a total life of 160 h, shown by the image superimposed on Figure 7. Locally, the fracture plane deviated with respect to the tensile axis. Traces of slip lines were evident immediately juxtaposed to the crack path and in some cases up to 20 mm away from the fracture. By eye, the density and orientation of slip lines varied between grains.

Evidence for stress relaxation over time under constrained loading was recorded from a plain cylindrical specimen subjected to a tensile strain of 1.5%, Figure 8. Immediately after the application of the linear strain-controlled ramp a bulk stress of 982 MPa was attained. Over the total 24 h period under test the resultant stress fell to 788 MPa, Figure 8 (top). However, the relaxation in stress was greatest during the earliest period of loading, falling to 859 MPa within the first 2 min, Figure 8 (bottom).

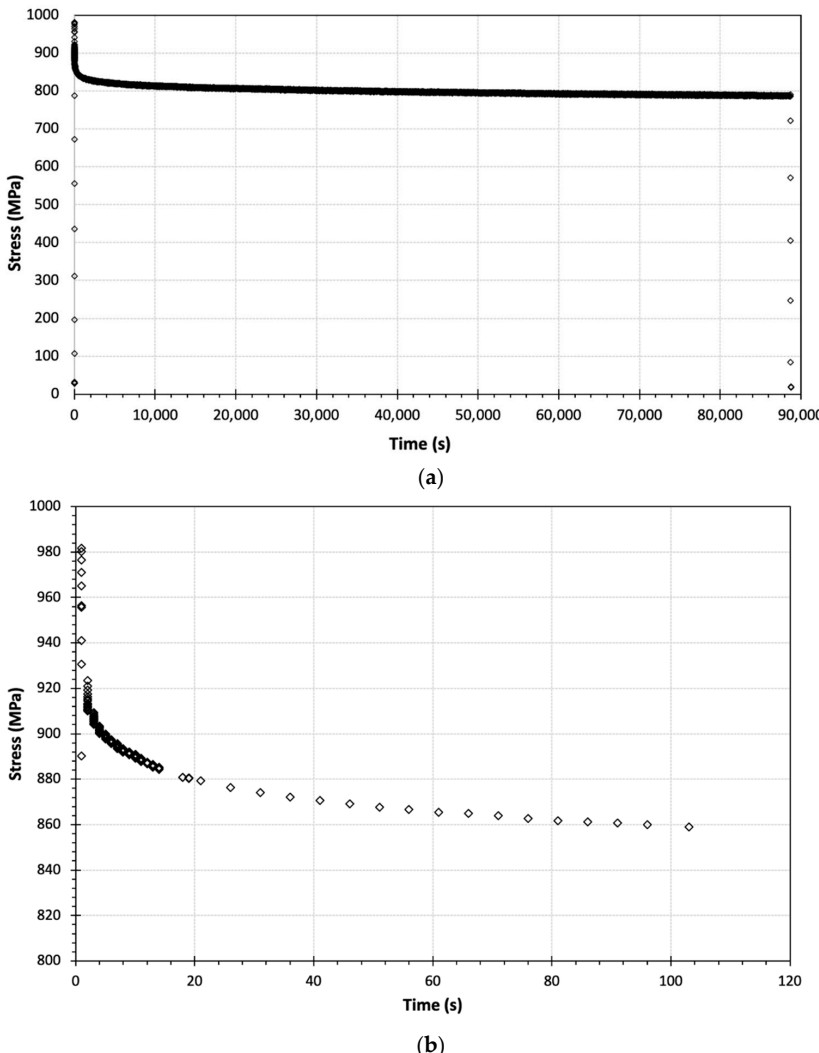

**Figure 8.** (**a**) Stress relaxation under a fixed strain of 1.5% during 24 h; (**b**) relaxation during initial 2 min.

## 3.2. Strain Controlled Fatigue Behaviour

Further evidence of constitutive behaviour was afforded by a matrix of strain-controlled fatigue tests. The fatigue data are illustrated in Figure 9 as a plot of peak applied strain against cycles to failure.

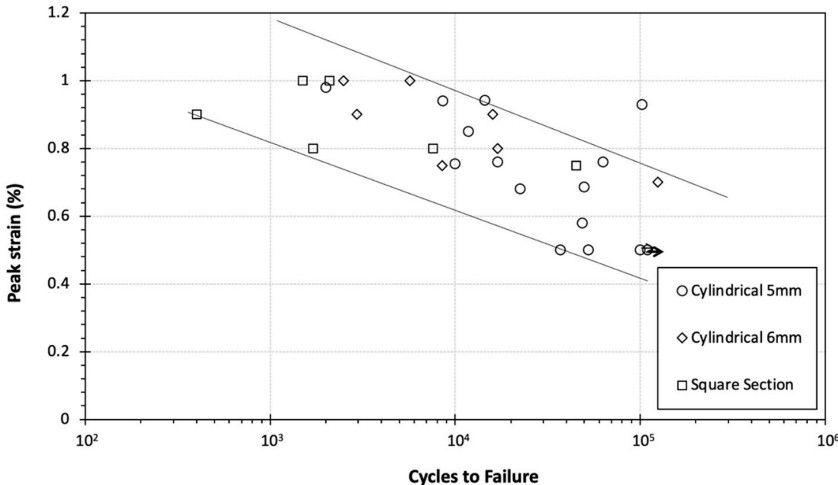

**Figure 9.** Strain controlled fatigue data.

The dataset combines the present results from the 6 mm diameter cylindrical specimen with those from a previous study conducted in this laboratory on the same LG685 material (using a 5 mm diameter cylindrical and 5 × 5 mm square section specimen designs) [22]. Trend lines are superimposed by eye to bound the majority of the database simply as a subjective method to emphasise the lack of sensitivity to sample design. At any given magnitude of applied peak strain the range in measured fatigue life almost extended to two orders of magnitude. This is further emphasised by the outlying point for a single test performed at 0.93%, which demonstrated a relatively long life to failure (slightly over $10^5$ cycles) compared to repeat tests at a similar condition. It can be assumed that the degree of scatter in fatigue life correlates to the variation in the number of grains sampled by each specimen and their specific crystallographic orientations.

Each specimen illustrated a bulk cyclic softening response throughout the course of the test, as exemplified by a specimen tested at $\varepsilon_{max}$ = 0.68%, Figure 10.

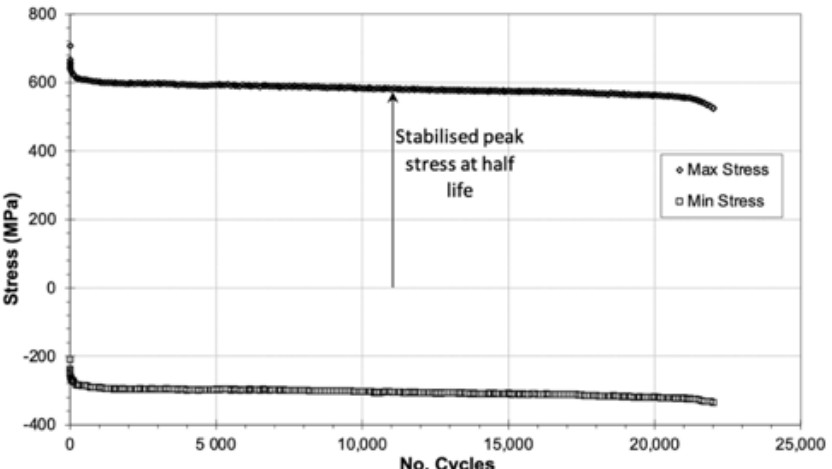

**Figure 10.** Bulk cyclic softening response under applied cyclic strain of 0.68%.

Individual stress–strain loops are plotted from this test at N = 1, 2, 8 and 500 in Figure 11. Monotonic yield in this specific specimen, as defined during the initial cycle (N = 1), was at approximately 500 MPa followed by a substantial degree of plasticity to form the open σ-ε loop. Subsequent cycles essentially appear to behave in an elastic manner; however, progressive cyclic softening was represented by a gradual reduction in both the peak and minimum resultant stress.

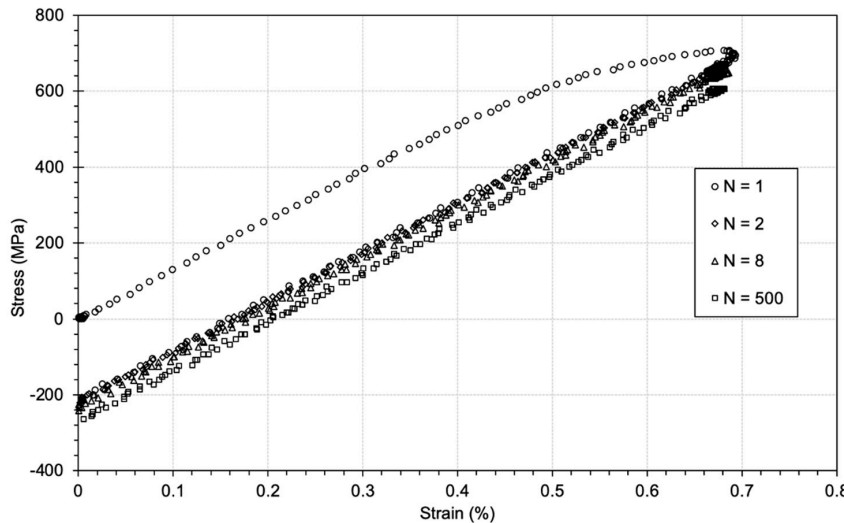

**Figure 11.** Constitutive behaviour during cycles 1, 2, 8 and 500.

Utilising measurements from the individual specimens, the stabilised peak cyclic stress at half-life (refer to Figure 10) was plotted as a function of the applied peak strain for each test to generate the cyclic stress strain curve (CSSC) for the LG685 alloy, Figure 12. When superimposed against the monotonic stress–strain curve (MSSC), represented by the first half cycle applied to the specimen loaded to 0.8% peak strain, the cyclic performance of the alloy was notably weaker.

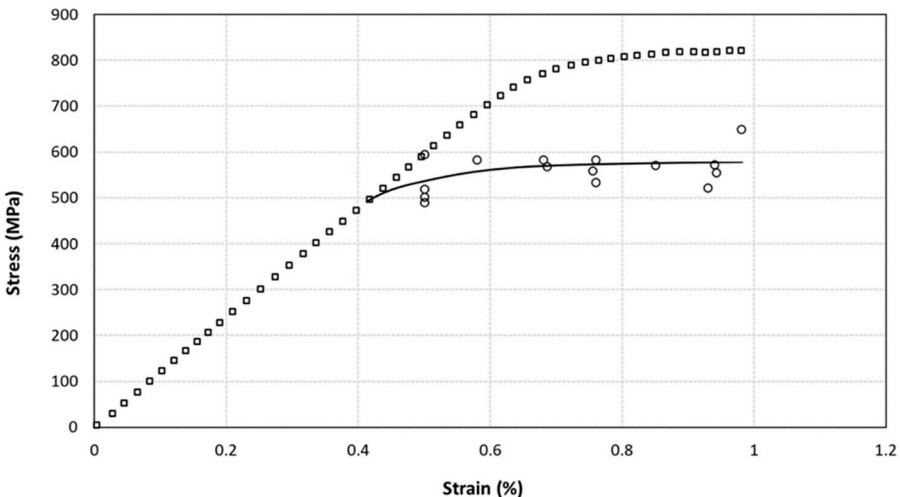

**Figure 12.** Monotonic and cyclic stress strain curves for LG685.

*3.3. Load Controlled Fatigue Behaviour*

Load controlled low cycle fatigue data from specimens subjected to cyclic and two-minute dwell waveforms are plotted in Figure 13. Mathematically calculated best fit power trendlines are superimposed on the two data sets. For applied peak stress conditions which generated fatigue lives of less than 20,000 cycles the alloy illustrated a weaker response under the dwell waveform (often described as a "dwell debit"). Around the N = 30,000 lifetime the two separate datasets appear to merge, corresponding to an applied stress condition of approximately 600 MPa.

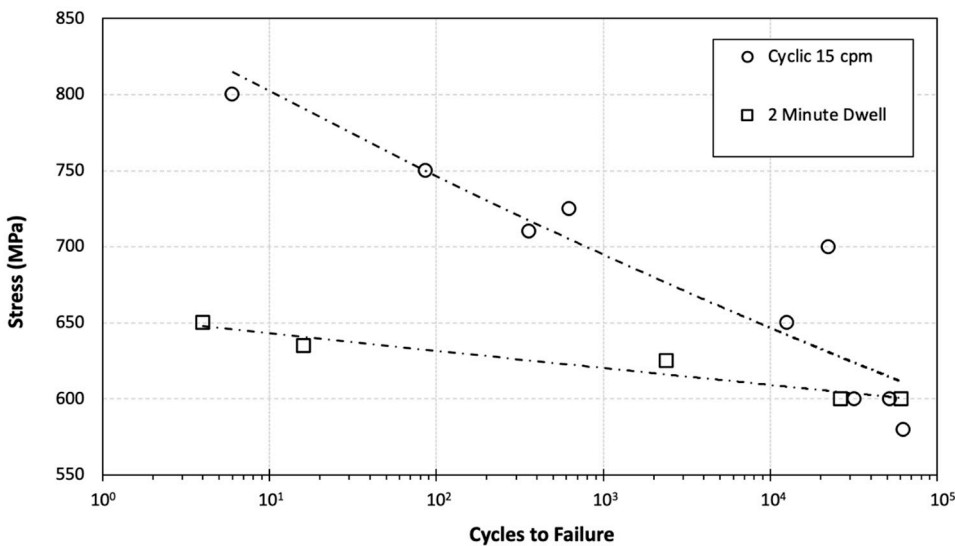

**Figure 13.** Load controlled fatigue data from cyclic and dwell testing.

*3.4. Failure Mechanism*

Fractography taken from both strain and load controlled cylindrical specimens exemplified the extremely crystallographic nature of the failure process, Figure 14. An example

of failure in a strain-controlled specimen, illustrates distinct, large scale, planar regions with well-defined intersecting boundaries. Some of these planes are orthogonal to each other and even lie parallel to the principal tensile loading axis. Planes dominated by high shear deformation illustrated outcropping linear slip traces on the fracture surface, whilst any quasi-cleavage facets formed orthogonal to the tensile axis were synonymous with radiating river markings. These fractographic forms have been previously documented in conventional titanium alloys [3]. Similar features were noted in the load control fatigue specimens.

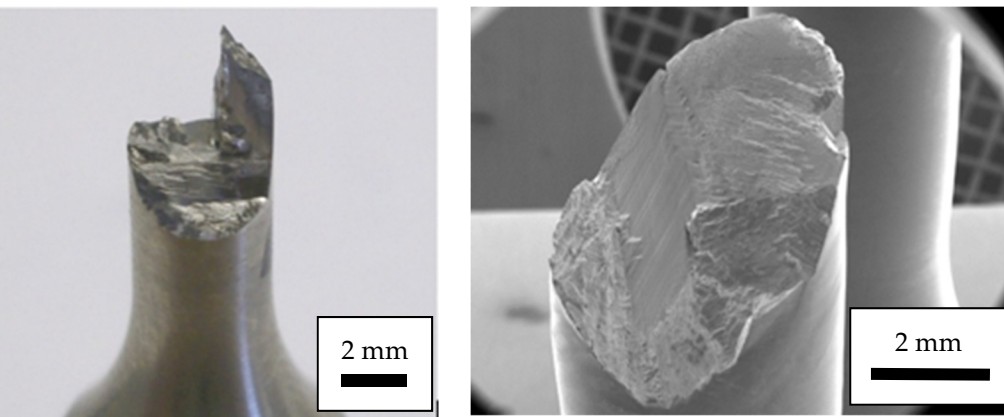

**Figure 14.** Macroscopic planar fractures generated under strain controlled LCF.

Occasionally fracture occurred on a single, inclined plane and fully extended across the cylindrical gauge section, Figure 15a. In this example from a stress relaxation test (constant strain = 1.5%) the failure clearly initiated sub-surface and subsequently grew in a progressive manner, evidenced by the radiating "river markings" and change in the surface reflectivity under optical light microscopy as the crack breaks free to atmosphere at the gauge periphery. At the epicentre of this fracture a distinct, featureless facet was observed at high magnification, Figure 15b.

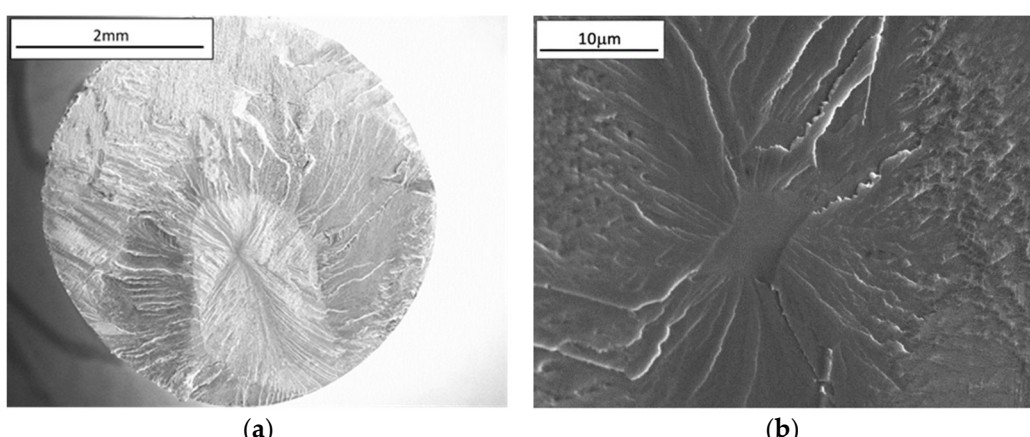

**Figure 15.** (**a**) Sub-surface initiation in a stress relaxation specimen; (**b**) high magnification view of the initiating facet (i.e., within the bounds of the area indicated in (**a**)).

## 4. Discussion

The experiments now described were deliberately designed to validate various aspects of cold dwell theory and modelling which invoke the initiation of quasi-cleavage facets as the key initiating failure mechanism. The observations and data presented, generated through the employment of an abnormally large-grained variant of the near α alloy Ti 685, are consistent with many previous works that have considered "cold dwell" and "dwell

sensitive fatigue" behaviour in standard forms of near α and α/β titanium alloys. It is pertinent to refer to both of these specific scenarios, i.e., under static and cyclic loading respectively, since the fundamental controlling factor for facet formation in each case is the time dependent accumulation of strain damage.

The fundamental requirement for the development of quasi-cleavage facets on basal (or near basal) planes lying orthogonal to the applied principal stress axis is the juxtaposition of strong and weak microstructural units. A schematic representation of this mechanism is reproduced in Figure 16 (taken from reference [13]).

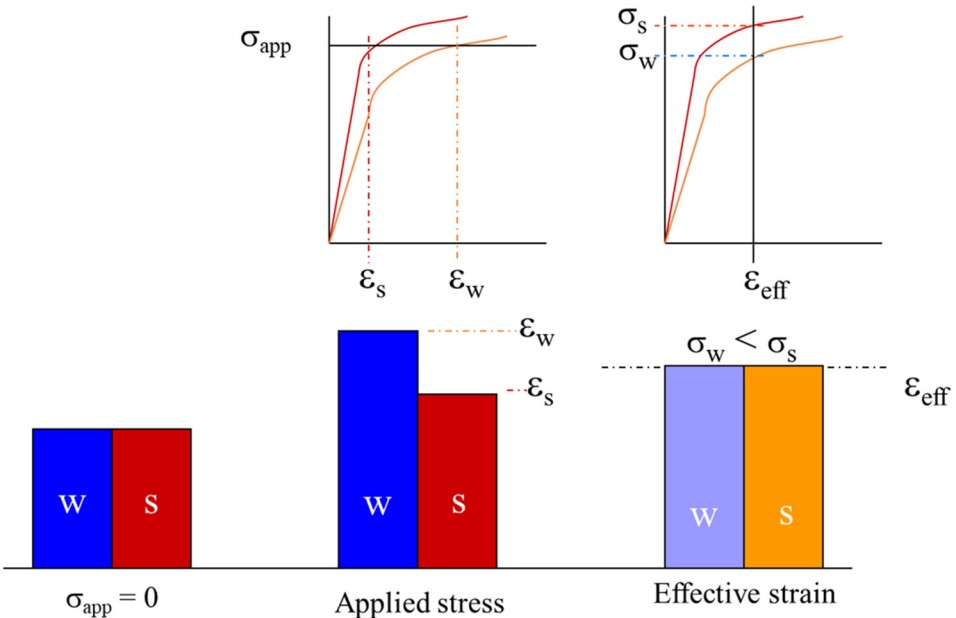

**Figure 16.** Two element model to describe stress redistribution between weak (W) and strong (S) adjacent grains [13].

The large grained LG685 variant studied here offered multiple examples of such circumstances in the various forms of mechanical test specimen employed. This occurred entirely fortuitously, since the specimens were extracted from the heat-treated block of material in a random fashion without prior knowledge of crystallographic orientations. However, observation and interpretation was clearly aided by the scale of the deformation and fracture features, together with the in-situ employment of strain gauges within individual grains.

Differences in the rate and absolute magnitudes of strain accumulation between grains were clearly illustrated in Figure 7. The strain–time relationship takes the basic form of a classical creep curve normally displayed by metals at elevated temperatures, with at least primary and secondary (or steady state) regimes evident [23]. Numerous reports of such static behaviour in titanium alloys have led to the descriptor "cold creep". A degree of elastic deformation was recovered during the episode of unloading mid-test, however, the moderate compressive loading during this 36 h transit was insufficient to recover any additional time dependent damage. This observation is highlighted to contrast with the studies of Mills et al. who detected dislocation reversal and associated stress relaxation during the offloading of room temperature creep experiments on a binary titanium Ti-Al alloy [24]. Indeed, whatever the status of dislocation structure immediately prior to the load reversal illustrated in Figure 7, upon reloading to the previous condition the rate of renewed strain accumulation was identical (emphasised in Figure 7 by the slope of the steady state data before and after unloading/reloading). If these two grains had been located in juxtaposition to each other, it is possible to imagine them acting as weak and strong grains in the context of the two-element stress redistribution model, Figure 16.

Regardless, it has offered a clear insight into the inhomogeneous deformation probably active within all conventional near α and α/β titanium alloys.

Under fatigue loading conventional titanium alloys demonstrate a degree of cyclic softening behaviour. Exemplary artefacts of this phenomenon were measured from the LG685 specimens tested under strain control (see Figures 8, 10 and 11). Through definition of the stabilised peak stress induced under strain control at half-life, the cyclic stress–strain curve was generated for the alloy (Figure 12). The alloy was notably weaker under cyclic loading when compared to the monotonic condition. The definition of the precise cyclic yield stress from the database of points available is subjective, but a value of approximately 500 MPa appears reasonable. This is some 200 MPa lower than that displayed by the example of monotonic yielding in the same figure. This raises two important points. Firstly, any computer based finite element model of titanium components subjected to a fatigue environment must employ the cyclic stress–strain curve and not the monotonic equivalent to describe constitutive behaviour to predict performance. Secondly, and highly pertinent to the definition of dwell sensitive alloy variants, it is only when subjected to applied peak stress conditions above the cyclic yield stress where any significant dwell debit in terms of the number of cycles measured to failure typically occurs. This was demonstrated here for the LG685 material (Figure 13) but also previously reported for Ti 834 [4].

Stress relaxation occurring under a fixed strain was quantified in Figure 8, with a drop of approximately 200 MPa measured by the stage where steady state relaxation was achieved. Notably, a sharp fall in the resultant stress of 111 MPa was invoked during the first 120 s of strain hold. It is argued that such relaxation could also occur during individual fatigue cycles that employ a dwell at peak condition, regardless of the duration of the hold. Indeed this has been noted during the present matrix of strain controlled fatigue experiments at relatively high peak strain conditions. Selected σ/ε fatigue loops are illustrated in Figure 17 for tests performed towards the extremes of the test matrix. Whilst the relatively low peak strain condition of 0.5% induces an essentially elastic response the test at 0.93% generates a significant degree of plasticity during the first cycle of loading. But notably, during the 1 s hold at peak strain within the 15 cpm trapezoidal "cyclic" waveform the resultant stress decreases by approximately 100 MPa. This raises a philosophical question over the selection of trapezoidal waveforms for the generation of purely cyclic fatigue data, for titanium alloys at least, since the shortest of holds imposed at peak strain (or stress) conditions appears to influence relaxation behaviour and could potentially affect the relative difference measured between "cyclic" and "dwell" fatigue lives to failure.

As illustrated above, if under strain control the resultant cyclic stresses may relax. However, if subjected to load control strain damage would accumulate in a time dependent fashion. With the very rapid relaxation effect illustrated by Figure 8 in mind, the employment of a two-minute hold at peak stress for many laboratory assessments of the dwell effect could be questioned in favour of a shorter period (say 30 s) in order to facilitate a more efficient test matrix. The rate and magnitude of strain accumulation during load-controlled dwell fatigue experiments have been shown to exceed that achieved under fast cycles [25] and is consistent with the time dependency inherent in the dislocation pile up models. Ultimately, the mechanism of failure does not vary between cyclic and dwell loading scenarios. Both involve planar slip damage concentrated in weak regions, leading to dislocation pile ups at the boundaries with strong regions, the initiation of quasi-cleavage facets in the strong phase (singular within a grain or clustered within MTRs, Figure 18) and finally fatigue crack propagation. The key distinction between the two loading scenarios is the reduced number of cycles required under dwell loading to initiate the facet(s). With the reliance on cyclic plasticity to encourage these dwell failures, the effects are not apparent at bulk applied stress levels below the cyclic yield stress where cyclic and dwell SN data merge on to a single trend.

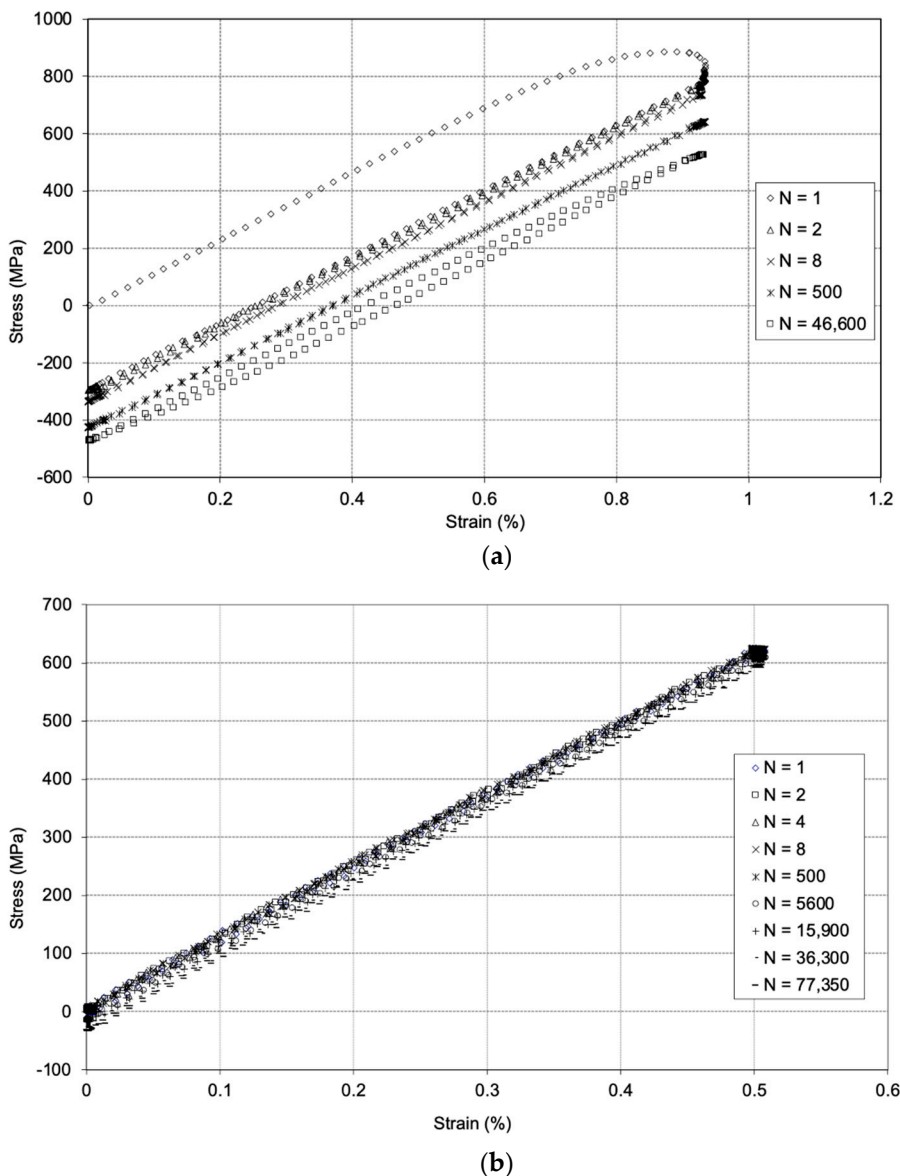

**Figure 17.** Selected hysteresis loops for strain-controlled fatigue tests performed towards the extremes of peak strain conditions: (**a**) 0.93% and (**b**) 0.5%.

Dwell fatigue failures in $\alpha/\beta$ and near $\alpha$ titanium alloys are often (but not exclusively) reported to initiate at sub-surface sites. In contrast with classical fatigue crack initiation models [26] both low and high cycle fatigue failures in these alloy variants have also been reported to initiate sub-surface [27,28]. The quasi-cleavage facet mechanism relies on the probabilistic distribution of weak and strong grains in juxtaposition to each other. It has to be remembered that traditional fractography of failed fatigue specimens, that has underpinned the understanding of facets over the past five decades, is reliant on features exposed on a single through section fracture plane. Recent in-situ characterisation of fatigue experiments on Ti-6Al-4V employing X-ray tomography has confirmed that multiple, sub-surface (and surface) initiation sites can form throughout the critically stressed gauge section of a plain cylindrical test specimen under fatigue loading [27]. The combination of the size of the initiating facet, local stress state and microstructure then dictates which of these embryonic cracks exceeds the fatigue crack growth threshold condition and subsequently leads to cyclic failure. Similar tomographic experiments under a dwell fatigue scenario should be encouraged.

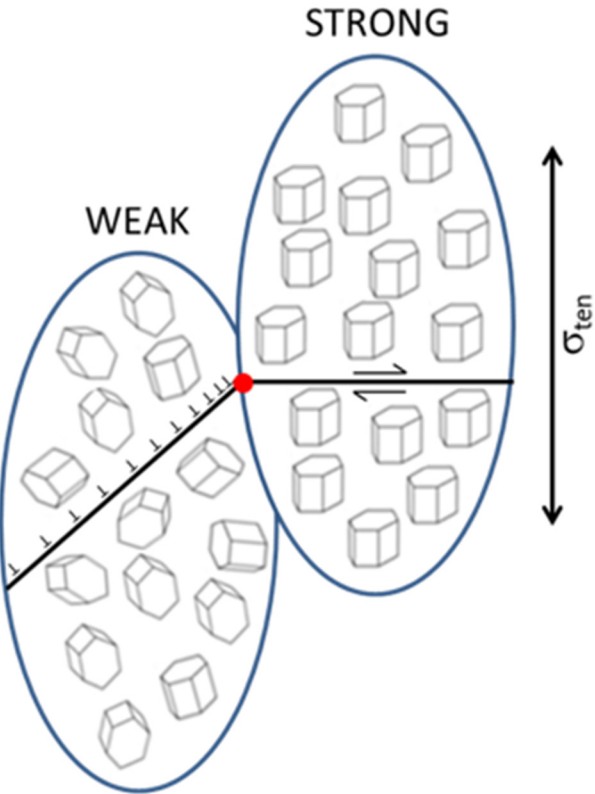

**Figure 18.** Proposed model to describe redistribution between "strong" and "weak" macrozones (MTRs) in finer scaled microstructures [12].

Quasi-cleavage facets can even occur under static "cold creep" loading, as exemplified in Figure 15. The absence of cyclic stressing during the formation of this example emphasises the greater importance of time dependent plasticity (over fatigue) to initiate such features.

The current fatigue data on the LG685 variant, whether generated under load or strain control, were subject to significant scatter. This is a direct consequence of the large grain size in relation to the relatively small specimen gauge section. In some cases, the specimen could sample a single grain. Therefore, the crystallographic orientation of the few grains sampled would dominate constitutive behaviour and subsequent fatigue life. The mechanical behaviour of hexagonal close packed titanium single crystals is known to be sensitive to orientation, in particular the inclination of the C axis to the tensile load [29]. The current variation in elastic modulus of individual grains was demonstrated by Table 2 and variations in yield properties can be inferred from the hardness data in Table 1. The LG685 fatigue fracture surfaces were highly distinctive. In selected specimens fracture was often restricted to a single, inclined crystallographic plane that spanned the entire cross section and in other cases fracture was even forced onto planes orientated parallel to the applied tensile stress, Figure 14. Macroscopic slip traces were clearly evident adjacent to the fracture surfaces, Figures 14 and 15 emphasising the planar nature of the plasticity.

Since the recognition of dwell sensitive fatigue behaviour in the late 1970s, the conventional form of Ti 685 was superseded for aero-engine applications by alternative alloys. In particular, two alloys adopted for compressor discs, Ti 834 and Ti 6242, have both illustrated dwell sensitivity during numerous laboratory studies and engineering service. In both materials, the presence of MTRs or "macrozones" has been identified as a major cause of anisotropic mechanical response, stress redistribution and fatigue failure. Over the same period, Ti-6Al-4V has been largely considered as a non-dwell sensitive alloy, serving as a popular choice for major fan disc components. However, it has long been recognised that thermo-mechanical processing of Ti 6/4 can also generate inhomogeneous microstruc-

tures. Indeed, the presence of a relatively prominent macrozone has been implicated as the source of a recent in-service fan disc failure [6]. The investigation report concluded that an improved understanding between cold dwell behaviour and microstructural condition was required for Ti 6/4, currently acting as the focus of continued International research efforts [30]. More recently, a modification to the Evans–Bache model to account for MTRs within a fine grained alloy variant has been proposed, Figure 18. Although a bespoke form of Ti 685 was the vehicle for the present study, it is argued that elements of the current findings are directly relevant to the fundamental understanding of cold dwell in Ti 6/4.

## 5. Conclusions

The following high-level conclusions are drawn from the present study:

1. The employment of a non-standard variant of Ti 685 with relatively large grain size proved successful for demonstrating inhomogeneous constitutive behaviour.
2. Distinct differences in elastic and plastic deformation properties were measured on the length scale of individual grains.
3. Important time dependent loading effects were characterised which assist our previous understanding of stress redistribution, quasi-cleavage facet formation and cold dwell mechanisms.
4. It is evident that any computer based finite element model of titanium components subjected to a fatigue environment must employ the cyclic stress–strain curve and not the monotonic equivalent to describe constitutive behaviour in order to predict performance.
5. A dwell fatigue debit was only observed when peak stress conditions were applied above the cyclic yield stress for this material.
6. In light of the predominant stress relaxation effect illustrated in the earliest stages of the applied dwell cycle, it is suggested that to generate a more efficient test matrix a maximum dwell hold time of 30 s could be employed instead of the 120 s often used.
7. At the same time, trapezoidal waveforms may not be the most suitable to generate "cyclic" fatigue data since at high applied stress (or strain) conditions relaxation effects are imposed during the short hold at peak stress.

These combined findings are highly relevant in light of the continued interest in cold dwell performance of safety critical titanium components in engineering service.

**Author Contributions:** Conceptualization, M.R.B.; methodology, E.E.S.; validation, E.E.S. and M.R.B.; formal analysis, E.E.S.; investigation, E.E.S.; data curation, E.E.S.; writing—original draft preparation, E.E.S. and M.R.B.; writing—review and editing, E.E.S. and M.R.B.; project administration, M.R.B.; funding acquisition, M.R.B. Both authors have read and agreed to the published version of the manuscript.

**Funding:** The research was sponsored by the Air Force Office of Scientific Research, Air Force Material Command, USAF, under grant number FA8655-04-1-3023. The US Government is authorized to reproduce and distribute reprints for Government purpose not with-standing any copyright notation thereon. The views and conclusions contained herein are those of the authors and should not be interpreted as necessarily representing the official policies or endorsements, either expressed or implied, of the Air Force Office of Scientific Research or the US Government. The supply and heat treatment of the LG685 material by Timet UK Research Labs, Witton, Birmingham is gratefully acknowledged.

**Data Availability Statement:** The raw/processed data required to reproduce these findings are available from the corresponding author upon reasonable request.

**Conflicts of Interest:** The authors declare no conflict of interest.

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
