# Peer review of "Novel Experimentation for the Validation of Mechanistic Models to Describe Cold Dwell Sensitivity in Titanium Alloys"

_metals, doi:10.3390/met11091456_

Round 1
Reviewer 1 Report
The present study proposed to measure the elastic and plastic deformation behavior of the Ti alloy. The topic is interesting and important for understanding the deformation behavior of Ti alloy. However, it is not clear what is the novelty in this manuscript. Please clarify the novelty here.
The introduction is not comprehensive. Please add more literature surveys to find the literature gap.
What is the advantage and limitation of this model?
The experimental detail is not sufficient to reproduce the data. Please update it.
Please add more information and the mechanism of the proposed model.
Author Response
- The present study proposed to measure the elastic and plastic deformation behavior of the Ti alloy. The topic is interesting and important for understanding the deformation behavior of Ti alloy. However, it is not clear what is the novelty in this manuscript. Please clarify the novelty here. With the exception of two papers previously published by the same team at Swansea, to our knowledge there are no other examples in the literature of mechanical assessment on this commercial near alpha titanium alloy in such a novel microstructural form. The bespoke material processing and associated experimental methods have allowed for the direct characterisation of inhomogeneous mechanical response relating to crystal orientation and micro-texture. This is near impossible to perform on the standard microstructural forms of titanium alloys without resorting to X-ray based techniques with the associated high cost and limited availability. An additional outcome is that the findings now raise interesting questions over the selection of trapezoidal forms of fatigue waveform to generate cyclic fatigue life data without the influence of material relaxation.
- The introduction is not comprehensive. Please add more literature surveys to find the literature gap. The paper was not intended as a review but instead serves as an incremental update to cold dwell linked issues through a self consistent empirical study. Notably one of the co-authors, MRB, has published over 50 papers relating to the mechanical behaviour of titanium alloys and cold dwell issues in particular. Specific references to a selection of those previous papers are provided, including a much cited review paper on the cold dwell phenomenon (the original reference number 9) to allow the uninitiated reader full access to the topic of cold dwell. However, additions to the Introduction now make reference to the most recent programme of cold dwell research sponsored by the US Department of Defense in collaboration with the USAF research labs and the major engine manufacturers – the Materials Affordability Initiative (MAI). Selected papers published from the results of an early work package and as presented to the most recent World Titanium Conference in 2019 have now been included in the Introduction together with three from parallel research groups working outside of that consortium.
- What is the advantage and limitation of this model?
and
- Please add more information and the mechanism of the proposed model. Models for quasi-cleavage facet formation based on stress redistribution between “weak” and “strong” grains have been proposed across a number of previous publications by Evans and Bache plus co-workers since the early 1990s. These have been widely accepted and cited amongst the International cold dwell community and selected works are referenced in the current paper. The present constitutive measurements add credence to those models and as the title suggests provide a measure of validation to previous theoretical ideas. The argument that stress redistribution can also act on a regional scale due to microstructurally textured regions (MTRs) or “macrozones” has been supported by the empirical studies under the MAI initiative (see publications from Ventkatesh and Glavicic et al). Again, the data published in this paper adds support. These points have been reiterated in the Introduction, Discussion and Conclusions sections of the present paper. It is also hoped that the physical data and understanding described will subsequently act as an input for others to interpret and implement into their computer based models. For example, the research conducted by the team at Imperial College (F. Dunne et al) could employ such findings to support crystal plasticity modelling.
- The experimental detail is not sufficient to reproduce the data. Please update it. As requested by a different reviewer, additional detail has been inserted around the selection and application of strain gauges plus testing standards are also referenced where appropriate. The material heat treatment is proprietary information which has been explained in the text.
Reviewer 2 Report
Comments for authors are in the attachment.

Author Response
- There are many new articles about high-strength titanium alloys (for example, Prof.Alexey Panin’s, Prof. Mykola Chausov’s, Prof. Varvara Romanova’s, etc.) in which claim that the system study of titanium alloys is effective at macro- (sample), meso-(conglomerates of grains) and microlevels (individual grain). In this article, the author develops a mixed approach, when the properties of grains(micro level), as well as the strength of material (macro level), are considered and compared together. In my opinion, this creates confusion in the presentation of the results. It is necessary to systematize the study of deformation processes at various scale levels, to find physical and mechanical connections between them. This will allow you to correctly study the material’s properties. The specific objective of the current research was to identify and quantify inhomogeneous mechanical behaviour in the large grain 685 variant that can then be assumed to occur in standard forms of the alloy (plus other near alpha, alpha/beta variants). In particular, to improve our understanding of the stress redistribution process that can occur between neighbouring grains and lead to quasi-cleavage facet formation. The focus was always on the localised deformation and fracture process that ensues rather than the understanding or prediction of bulk alloy properties. We have attempted to place greatest focus on the microscale behaviour, but in order to describe and explain some of the experimental procedures it was necessary to supplement this with information relating to bulk specimens (both under static and fatigue loading scenarios).
- Figure 1 shows well-known diagrams. Diagrams are given in the Introduction, which is not traditional for scientific journals. I suggest deleting it. The diagram in question is essential to place the current research into context. To satisfy your preference not to insert it in the Introduction it is now simply referenced at that stage, but reproduced later during the early stages of the Discussion section.
- To measure the deformations of individual grains of the material the small-size strain gauges were used. But in the article does not indicate their type, deformation recording rate, measurement base, measurement accuracy. The absence of these data makes it impossible to assess the correctness of the measurements. It is necessary to compare the base of measurements of the strain gage and the grain size and justify the correct choice of these strain gages. t is not clear how the strain gauges were attached to the samples. What kind of glue was used? This is also very important for the correctness of data recording. Detail of the types of strain gauges used, original manufacturer (thus allowing access to full technical specifications), how they were applied and sampling rates have now been included.
- In fig. 6 shows the stress-strain curve measured from a flat plate LG685 specimen. It is indicated that the properties of this material are different from the standard (fine-grained). I propose to indicate in how many % this properties are different. In addition, this material has a very low deformation value of up to 2.5%. How correct speak about “cold creep” for such a low plastic material? I recommend describing this diagram in more detail.
|
Property at 20oC |
Value |
|
0.2% Yield Strength |
900MPa |
|
UTS |
1030MPa |
|
Elastic Modulus |
125GPa |
The data in the Table above are taken from the original reference 14, so a comment is now inserted in the Results to state that the LG variant has an inferior strength by approximately 20% when compared to standard 685 variants. We have also added comments in the Discussion to recognise the bulk strain at failure measured during the tensile test correlates closely with the maximum creep strain accumulated at failure in the “weak grain” of the cold creep test. These alloys are notorious for demonstrating creep at room temperature if loaded beyond yield irrespective of bulk tensile ductility.
- I recommend you carefully study the sample loading system in the testing machine. I'm not sure if the one shown in fig. 7 relaxation is the result of grain deformation. Perhaps this is relaxation in the loading system of the machine. It is necessary to show not only this data, but also the value of the load. Changing it during these tests will be very informative. I propose to rebuild this diagram in the form of the recommended ASMT. This will allow all significant dates to be analysed. We are not entirely sure what the reviewer is questioning. The test described in Figure 7 was performed under machine load control on a fully calibrated servo-hydraulic test rig, programmed to apply a static stress to the specimen at 600MPa (i.e. nominal load = 30kN, subject to precise specimen dimensions). The applied load remained constant throughout the course of the test whilst resultant strain was measured directly off the face of the specimen via strain gauges (fixed within individual grains). In our experience, plotting strain as a function of time is the standard method to demonstrate “creep” behaviour, as per the classical text original referenced as #16.
- "Сold creep" of this material raises doubts for me. This effect should be shown on the basis of images of grains using optical or scanning microscopy. Referring to Figure 7, visual evidence of permanent “creep” deformation is clearly illustrated within the different grains. Planar slip features have been highlighted as evidence of micro-scale creep plasticity. More detailed microscopy (in the form of TEM etc) was not generated as part of this research. In fact, one of the novel aspects of the present research is the ability to recognise such localised, plastic deformation by eye due to the scale of the grains assessed.
- In fig. 12 the points of monotonic and cyclic stress strain curves for LG685 need to be marked. Apologies, a legend with the MSSC & CSSC data sets now defined has been included.
- Please, on fig. 13 draw the shapes of the loading cycles. Also need to indicate how the loading frequency has changed. The waveform “shapes” and “frequency” are fully described in the text within the experimental section, however, a schematic representation is now also included.
- Micromechanisms of failure shown in figs. 14, 15 needs to be described in more detail. Distinguishing features noted on the neighbouring shear and facet planes are now described. More details on these typical features and the mechanism of their formation are referenced back to a previous paper (original ref #3).
- In the section "Discussion" it is necessary to systematize the results. New formulas, models, physical laws, tables, diagrams are very necessary in this section. Following your suggestion, the original Figure 1 has now been incorporated into the Discussion section as a link to the original rationale for this research and to correlate to the new results now presented. We contend the current text accurately describes the key outcomes from the Results section, leading to a subsequent list of key conclusions.
- The conclusions are written very superficially. There are no numerical results in conclusions. It is not shown what results new for world science and practice were obtained in this article. Also, the annotation needs to be clarified. It also does not reflect the novelty of the article results. The Conclusions section has been revised to include a list of key findings and a final statement that addresses the relevance of the outcomes to ongoing engineering issues concerning cold dwell performance in aero-engine components.
Reviewer 3 Report
In this paper, the authors designed a novel experimentation for the validation of mechanistic models to describe cold dwell sensitivity in titanium alloy. Some interesting results have been reported to demonstrate significant variations in elastic and plastic properties between grains and emphasise the role of time dependent strain accumulation. However, there are some considerations by the referee that have to be addressed before the work could be recommended for publication.
1、The references in the Introduction is outdated and the latest references in recent five years should be discussed.
2、 The authors should provide the rolling direction (RD), transverse direction (TD) and normal direction(ND) in the Figure 2.
3、How the authors get the value of yield stress? According to the offset of 0.2%, the value of 600 MPa seems wrong. In addition, the error bar in the Table 1 and Table 2 is absent.
4、Figure 8 could not shows the variation of stress along times well at the stabilization stage, whose vertical coordinates should be changed from 600 to 1000 MPa?
5、In the Figure 15, the authors should mark the position of Figure 15b in the Figure 15a.
6、The conclusion is sample and cannot reflect the work of this study, which should be improved.
Author Response
- The references in the Introduction is outdated and the latest references in recent five years should be discussed. The Introduction now makes greater reference to the most recent programme of cold dwell research sponsored by the US Department of Defense in collaboration with the USAF research labs and the major engine manufacturers – the Materials Affordability Initiative (MAI – Work Packages 18 and 24). Papers published as the results of WP18 and as presented to the most recent World Titanium Conference in 2019 have now been included in the Introduction. Other references to recent cold dwell research publications are also inserted.
- The authors should provide the rolling direction (RD), transverse direction (TD) and normal direction(ND) in the Figure 2. The stock material did not experience thermo-mechanical rolling or forging processes that could introduce either hot or cold work where such notations would then be significant to the understanding of mechanical properties. To our knowledge the volume of material was extracted from the mid to late stage of ingot processing. Orientations relative to the ingot process were deemed proprietary by the material sponsor.
- How the authors get the value of yield stress? According to the offset of 0.2%, the value of 600 MPa seems wrong. In addition, the error bar in the Table 1 and Table 2 is absent. We have quoted yield strength at the approximate deviation point from the proportional elastic response (not a 0.2% proof). To provide greater clarity, this is now described more clearly in the text. The data quoted in Tables 1 and 2 were taken as single measurements relating to independent, isolated grains (not averages). The purpose was to emphasise the differences between grains and not to measure average properties to a given accuracy.
- Figure 8 could not shows the variation of stress along times well at the stabilization stage, whose vertical coordinates should be changed from 600 to 1000 MPa? We have not altered Figure 8a since we want to illustrate the data points during loading from zero, however, to emphasise the relaxation effect and resolve thae data more easily we have taken your advice and actually decreased the Y axis range between 800 and 1000MPa.
- In the Figure 15, the authors should mark the position of Figure 15b in the Figure 15a. Advice taken, Figure revised.
- The conclusion is sample and cannot reflect the work of this study, which should be improved. The Conclusions section has been revised to include a list of key findings and a final statement that addresses the relevance of the outcomes to ongoing engineering issues concerning cold dwell performance in aero-engine components.
Reviewer 4 Report
Dear Authors,
I am suggesting to major review. The paper is very interesting and covers important topic related to the experimental mechanics. However, several issue should be raised in revised version.
- Line 92 is mentioned: „Post heat treatment” - Please describe this heat treatment in detail
- 3.2 – Please provide more details in which way was controlled strain level during experiment…using extensometer? Does follow this test for any standard? Which failure criterion was used?
- 9 - It is better to plot the Coffin-Manson diagram instead of e-N curve. Moreover, a significant data scatter is noticeable – so – for cyclic testing was better to select more specimens for each load level (decreasing number of different types of specimens – what was the purpose of using different geometries?)
- 13 How many specimens were repeated for each load level? If one it is insufficient
- Section Failure Mechanism should be extended, including more images from SEM for different load levels
- 11 – is mentioned constitutive data modelling – please add any parameters of data fitting and add more results from hysteresis loop analysis, including the cyclic response of material for different load levels.
Author Response
- Line 92 is mentioned: "Post heat treatment” - Please describe this heat treatment in detail. The material heat treatment is proprietary information which has been explained in the text.
- 3.2 Please provide more details in which way was controlled strain level during experiment…using extensometer? Does follow this test for any standard? Which failure criterion was used? Strain control fatigue experiments were conducted according to BS7270 which also defines the criterion for failure (i.e. 10% drop from stabilised stress). Details now included.
- 9 - It is better to plot the Coffin-Manson diagram instead of e-N curve. Moreover, a significant data scatter is noticeable – so – for cyclic testing was better to select more specimens for each load level (decreasing number of different types of specimens – what was the purpose of using different geometries?). Coffin-Manson analysis assumes homogeneous mechanical response from the material under assessment. Representing the data in the form of Coffin-Manson is not appropriate here because of the unique bulk properties of each specimen and the inhomogeneous behaviour occurring in critically loaded, independent grains. Specimen specific data would be required to partition elastic-plastic performance which would not be representative of the nominal alloy. Different specimen geometries were deliberately selected here for different purposes of the study (e.g. flat plate for grain deformation measurements, plain cylindrical for fatigue life assessments). We then incorporated different specimen geometries used in previous studies in an attempt to expand the database and emphasise the effects of different microstructure sampled on fatigue life. Despite the numerous designs employed we believe the results are consistent.
- 13 How many specimens were repeated for each load level? If one it is insufficient. the number of tests were severely constrained by the volume of material made available by the sponsors. While we agree with your comments on multiple data per applied stress/strain condition (at least for the assessment of a standard homogeneous material) this was not feasible during the present study due to limitations in the volume of material available to us. Nevertheless, we believe the rationale behind our study, i.e. to emphasise variable response according to microstructure/crystallographic orientation, was achieved.
- Section Failure Mechanism should be extended, including more images from SEM for different load levels. We can confirm there was no trend in fracture behaviour with different applied stress/strain levels, therefore we have presented “representative” examples in this paper to most efficiently utilise publishing space. Greater detail on the fracture surface appearances has now been included with reference to previous papers offering more examples.
- 11 – is mentioned constitutive data modelling – please add any parameters of data fitting in Fig. 9 the trend lines are quoted as being drawn “by eye” to reference the scatter. The lines superimposed “by eye” in Figure 9 have no analytical basis, as described in the text these were drawn in a purely subjective fashion to indicate the degree of scatter in the measured lives across the range of applied stresses. In Figure 13, power fit trend lines as calculated using MS Excel are now superimposed for the 2 data sets with a clarification added to the text. Additional results from hysteresis loop analysis have been appended in a new figure. Although s-e behaviour was highly variable between specimens, a general trend in loop development was noted according to applied peak strain condition (ranging form elastic to plastic behaviour). Examples of extreme cases at 0.5 and 0.93% peak strain have now been inserted together with descriptions of their behaviour.
Round 2
Reviewer 2 Report
Comments are in the attachment.

Author Response
The authors have on recommendation revised the introduction of the manuscript which adds additional detail which references http://dx.doi.org/10.1016/j.msea.2017.05.029.
Reviewer 4 Report
The paper can be now accepted
Author Response
Many thanks for your feedback.